# Qualitative process evaluation of the EmpaTeach intervention to reduce teacher violence in schools in Nyarugusu Refugee Camp, Tanzania

Mtumwa Bakari,[1] Elizabeth H Shayo [ID],[2] Vivien Barongo,[1] Zenais Kiwale,[1] Camilla Fabbri,[3] Ellen Turner,[3] Emily Eldred,[3] Godfrey M Mubyazi,[1] Katherine Rodrigues,[4] Karen Devries [ID] [3]

[1]National Institute for Medical Research, Dar es Salaam, Tanzania
[2]Health Systems, Policy and Translational Research Section, National Institute for Medical Research, Dar es Salaam, Tanzania
[3]Global Health and Development, London School of Hygiene and Tropical Medicine, London, UK
[4]Research and Innovation Department, International Rescue Committee, New York, New York, USA

**Correspondence to**
Elizabeth H Shayo;
bshayo@yahoo.com

## ABSTRACT

**Objective** We explored the experiences and perceptions of school staff and students with the EmpaTeach intervention to prevent teachers' violence against school students.

**Design** This qualitative study involved in-depth interviews with 58 and 39 participants at midline and endline, respectively, with Burundian and Congolese intervention schools in Nyarugusu refugee camp. They comprised three education coordinators of primary and secondary schools, 29 EmpaTeach intervention coordinators, 14 stakeholders including headteachers and discipline teachers, 25 classroom teachers and 26 students. Thematic analysis was used to develop codes by examining the content of quotes to capture key themes in line with the key elements of the programme theory.

**Results** Coordinators and teachers widely reported positive experiences with the EmpaTeach programme. The intervention sessions enabled teachers to reflect on their own values and experiences of corporal punishment and equipped them with useful and acceptable classroom management and alternative discipline strategies. Teachers adopted the use of counselling, praise and reward, and joint discussions with students and parents. On the other hand, several teachers reported persistent use of corporal punishment which they attributed to children's (mis)behaviours and strong beliefs that beating was a positive approach to disciplining students.

**Conclusion** The majority of coordinators and teachers widely accepted the EmpaTeach intervention as it offered useful and relevant knowledge and skills on alternative disciplinary methods. Students noticed some positive changes on the way they were being disciplined by teachers, where non-violent methods were used. Further research is needed to understand how violence prevention interventions can successfully lead to reductions in violence in fragile settings.

**Trial registration number** NCT03745573.

## STRENGTHS AND LIMITATIONS OF THIS STUDY

⇒ The qualitative methods used to evaluate the intervention helped to get detailed information which could have been difficult to capture quantitatively.
⇒ Using of data collectors conversant with Burundian and Congolese language facilitated gathering rich data because of smooth communication.
⇒ The study recruited few participants, hence it was difficult to generalise findings to the whole population.
⇒ Adding and/or replacing some of the participants may have limited our ability to compare perspectives and insights over time.

is one of the most visible forms of violence against children, and can be perpetrated by teachers and other school staff in the playground, classroom or any place within school premises.[1] Physical violence from teachers can include corporal punishment aimed at imparting discipline and sanctioning misbehaviours.[2 3] In Tanzania, the use of corporal punishment is rooted in the belief that it plays a key role in shaping students to conform to society's rules and norms and steer them towards achieving good academic outcomes.[4] Yet, severe physical punishment can result in physical harm and affect children's mental health.[5] Consequently, the elimination of all forms of violence against children, including corporal punishment, is included in Sustainable Development Goals 5 and 16.[6]

Several interventions have been conducted in non-humanitarian settings aiming to prevent violence from teachers to students. A cluster randomised controlled trial evaluated the Irie Classroom Toolbox in Jamaica, and showed that teacher training on classroom behaviour management, using a combination of workshop and in-class support sessions, helped to reduce teachers' perpetration of

## INTRODUCTION

Violence against children is a global public health concern, whose consequences can harm children, families, communities and nations.[1] Physical violence in school settings

physical violence against children.[7] The Good School Toolkit (GST) in Uganda, also found to be effective at reducing teacher violence,[8] intervened at the whole-school level and focused on changing schools' operational culture.[9] The intervention employed behaviour change strategies for schoolteachers, administration and students, aiming to encourage empathy by reflecting on experiences of violence, imparting new knowledge on alternative non-violent discipline, and creating opportunities for practising new skills. In addition to reducing physical violence from school staff, qualitative research suggested that the intervention strengthened student–teacher relationships, improved students' voices and reduced fear of teachers.[10] There is suggestive evidence that the Interaction Competencies with Children for Teachers (ICC-T) was also effective at reducing teacher violence through strategies to improve teacher–student interaction, including reflections on maltreatment, alternative discipline strategies, and support for students.[11]

While these trial findings are encouraging and show that it is possible to reduce teacher violence, no study to date has focused on teacher violence in humanitarian settings. The risk of violence in humanitarian settings is exacerbated by a variety of factors at each stage of the social–ecological model;[12 13] specifically students in refugee camps are likely to suffer from the double burden of corporal punishment in school and the trauma of having been uprooted from home. To address this evidence gap, the Preventing Violence against Children in Schools study was designed to evaluate the EmpaTeach intervention to reduce teacher violence in Nyarugusu Refugee Camp in Tanzania. EmpaTeach is a behavioural intervention for teachers aimed at reducing their use of corporal punishment in the classroom, designed and implemented by the International Rescue Committee (IRC) in collaboration with the Behavioral Insights Team. The study consisted of a cluster randomised controlled trial to assess the effectiveness of the intervention, and embedded qualitative and quantitative process evaluations focused on assessing intervention implementation and mechanisms of action.[14 15] The trial found no evidence that the intervention reduced student self-reported experiences of physical violence from teachers in schools, and similarly found no effect on any of the secondary outcomes such as students' experiences of emotional violence from teachers, student depression symptoms and school attendance. However, additional analyses of the trial data showed that the intervention improved some intermediate outcomes for teachers, such as using alternative discipline strategies and lowered their acceptability of violence; teachers in intervention schools also reported improved self-regulation compared with those in control schools.[15] In this paper, we aim to explain these trial findings and provide further evidence on the experiences and perceptions of the EmpaTeach intervention among school staff and students and its perceived effects in line with the programme theory.

## METHODS

### Study setting

The study was conducted in Nyarugusu Refugee Camp in Kigoma region, Tanzania. The camp was established in 1996 to host 80 000 refugees from the Democratic Republic of the Congo. Between April and October 2015, the camp expanded to host an additional 70 000 Burundian refugees. The latest United Nations Human Rights Council figures estimate the total population of the camp to be 150 000.[16] Tanzania's Refugees Act No. 9 of 1998 stipulates that refugees are not allowed to engage in formal employment and restricts the movement of refugees after being settled in a designated area.[17] Consequently, the camp's population largely relies on relief food supplies and commodities from non-governmental organisations and United Nations agencies active in the camp.

The United Nations High Commission for Refugees and the Tanzania Ministry of Home Affairs run and administer the camp. Meanwhile, the IRC is responsible for all educational activities in the camp, in addition to providing gender-based violence support services. The camp population is generally divided by country of origin. There are 27 schools in the camp, which use either a Congolese or Burundian curriculum. These schools face numerous challenges, such as poor teacher attendance, shortages of qualified teachers, and inadequate infrastructures leading to difficulties in implementing quality education.

### EmpaTeach intervention

EmpaTeach is a behavioural intervention that aimed to reduce and prevent teacher's use of corporal punishment in schools. Through a series of 12 peer-guided group sessions, the intervention sought to shift teachers' attitudes and behaviours leveraging approaches from cognitive behavioural therapy. The weekly group sessions focused on (1) teaching participants positive classroom management and alternative discipline practices by engaging teachers in reflection and planning exercises and role plays that allowed them to practice these newly acquired methods. (2) Group activities were designed to develop teachers' self-regulation, self-efficacy, openness to change and empathy towards others to facilitate teachers' adoption of the new strategies and ultimately improve teacher well-being and ability to effectively manage students. (3) The group setting, and social support offered by peers were supposed to sustain teachers throughout the process of change and promote the adoption of new norms less accepting of violence.

The content of the EmpaTeach intervention differs from other trialled school violence prevention interventions for its specific focus on self-control and impulsive use of violence, in addition to promoting teachers' use of alternative discipline strategies without explicitly discouraging the application of corporal punishment. The intervention implementation and content are described in further detail elsewhere.[15] Box 1 describes the EmpaTeach strategies for teachers.

**Box 1 EmpaTeach recommended strategies**

**Alternative discipline methods**
⇒ Disciplinary (move closer, lower voice, sudden silence, change seats, polite pose, clapping, calling headteacher, calling parents, stay after schools and apologise) and praise practices (cheering, singing, additional homework and student of the week).

**Classroom management strategies**
⇒ General classroom management and teaching practices (lesson objectives, feedback, opportunities to practice, questioning techniques, classroom rules and use of space).

**Enhancing teachers' well-being**
⇒ Stress and emotional management techniques (count to 10, recognise triggers, positive sentence to self, take a quick walk, reading story books and praying).

**Table 1** Research participants

| Categories | Congolese | | Burundians | |
| --- | --- | --- | --- | --- |
| | Midline | Endline | Midline | Endline |
| EmpaTeach coordinators | 8 | 7 | 8 | 6 |
| Education coordinators | 2 | – | 1 | – |
| Headteachers | 4 | – | 4 | – |
| Discipline teachers | 3 | – | 3 | – |
| Class teachers | 7 | 8 | 5 | 5 |
| Students | 6 | 6 | 7 | 7 |
| *Total* | *30* | *21* | *28* | *18* |

EmpaTeach coordinators were the teachers who acted as group coordinators during intervention sessions.
Camp coordinators were overall coordinators for primary and secondary Burundian and Congolese schools.
Discipline teachers and head teachers fall under the group of stakeholders.
Class teachers were regular teachers.

### Study design

Qualitative data were collected at three time-points: baseline before the intervention in January to February 2019, midline soon after the 10 week of intervention from June to December 2019 and endline conducted 6 months after the end of the intervention from September 2020.

### Patients and public involvement in research

Patients and the public were not involved in study design. Teachers, coordinators and students from the schools that received EmpaTeach intervention were involved in the interviews as study participants.

### Sampling and recruitment of study participants

Participants were purposively selected based on the level of violence experienced in Congolese and Burundian primary and secondary schools in the intervention arm and were interviewed at baseline, midline and endline. They include teachers, students, and EmpaTeach intervention coordinators. Notably, some stakeholders such as education coordinators, discipline teachers and headteachers were interviewed once at midline. The camp education coordinator facilitated the recruitment. In this paper, we draw data mainly from the midline and endline due to the focus on changes deriving from the intervention.

A total of 58 and 39 participants were interviewed at the midline and endline phases, respectively (table 1). These participants comprised three education coordinators for primary and secondary schools, 29 EmpaTeach coordinators, 14 stakeholders including headteachers and discipline teachers, 25 classroom teachers and 26 students. Some participants who were unavailable at follow-up rounds of data collection were replaced by participants with similar characteristics, while some of the participants were interviewed twice including five coordinators, three teachers and three students. There was no refusal.

### Data collection procedures

Data were collected via face-to-face in-depth interviews. Interviewers included males and females with bachelor degrees in Social Sciences who were familiar with Kirundi and Congolese Swahili. Having experience in conducting qualitative work, they received training on research methods, techniques for working with children, and on child protection referral procedures. They piloted the tools before the actual fieldwork.

Interview guides focused on topics such as experience of teaching, emotional self-awareness and growth mindset, discipline and violence in school experiences, reflections on perceived changes in schools following the EmpaTeach intervention and any challenges during EmpaTeach implementation. These interviews were conducted in Congolese Swahili or Kirundi, audio recorded and transcribed verbatim. Field notes were also taken during the interviews. A private place for interviews was sought within the schools to ensure freedom of expression. Participants' consent to participate was given with knowledge that disclosures of abuse may be passed on to child protection officers. Saturation was observed and so interviews stopped when no new information was coming from the interviewees. For example, when asking about alternative disciplinary methods or rewarding mechanisms, similar information was reported repetitively, indicating that the saturation point was achieved. Each interview lasted between 30 and 45 min.

### Data management and analysis

Senior researchers re-read the transcripts and compared them with audio recordings to familiarise themselves with the data. Data analysis was thematic in identifying themes and patterns, by examining the underlying ideas, assumptions and conceptualisations on the use or non-use of alternative disciplinary methods.[18] Analysis involved developing codes through examining the content of each sentence or sequence of texts by two coders to capture the predefined key themes in each section of the interview

guide using excel spreadsheets. Codes were then sorted into main themes and subthemes, focusing on key aspects relevant to this study, including aspects of the EmpaTeach intervention such as behavioural changes from practising alternative disciplinary methods; changes in the general school environment; different techniques in classroom management; improved well-being through enhancing teacher–teacher and teacher–student relationships; challenges encountered; and positive and negative outcomes among students consistent with the components of the programme theory.

### Ethical considerations

Permission to conduct the study was obtained from the Tanzania Ministry of Home Affairs, the IRC and the camp authorities. Participants offered informed consent, and the voluntary nature of their participation was emphasised so that they could withdraw from the study at any time if they wished. For students under 18 years, headteachers, who stood on behalf of parents/guardians in their respective schools, provided initial written consent for their participation, and students offered their written assent. Study participants over 18 years provided written informed consent. Privacy, anonymity and confidentiality were maintained throughout the study, except in cases when the study team came across disclosures of child abuse raising immediate protection concerns. Participants were informed during the consent process about conditions under which confidentiality would be breached so that they could retain control over what they disclosed. We confirm that all the methods used in this study were in accordance with relevant ethical guidelines and regulations.

### RESULTS

We examine participants' perspectives of change in their schools and their experiences with the intervention in relation to the three components of the EmpaTeach intervention programme theory: (1) teachers practice EmpaTeach strategies; (2) teachers use emotional regulation and well-being techniques and (3) teachers feel supported in their process of behavioural change.

### Intervention implementation

Study participants reported the intervention to be successfully implemented. However, teachers and coordinators reported contextual challenges that affected their attendance. These included difficulties in balancing coordination work alongside teaching responsibilities as one of the coordinators reported:

> To be a coordinator and a teacher at the same time was not easy, because of our situation or environment, sometimes you prepare a lesson, and you do not find the ones to teach at the proper time. It is tiresome but if you work hard, you can shoulder both responsibilities.(Coordinator, midline)

The rainy season also interfered with the timely and effective delivery of intervention activities. There were disruptions to intervention implementation caused by a temporary firing and rehiring of new teachers which contributed to the lowering morale among teachers and may have affected both attendance and internalising some of the techniques taught.

### Teachers practice alternative disciplinary and classroom management strategies

Study participants, who were class teachers and EmpaTeach coordinators, mentioned strategies covered during the intervention sessions as part of EmpaTeach that they found useful. These strategies included providing advice and counselling students, using non-abusive language, participatory development of class rules and regulations, use of praise and reward and classroom management strategies.

> We learned some techniques that are positive including non-corporal punishment methods, rewarding students who perform well in the class and knowing how to control students in a classroom. (Teacher, endline).

Teachers described applying alternative discipline strategies that they had learnt during the intervention sessions. One of the teachers explained how in the past he used to stop teaching and start caning students if they misbehaved in class but after EmpaTeach such behaviour changed. This was substantiated by the following quote.

> In the past when were some misunderstandings between a teacher and a student, the latter used to be threatened and beaten. But soon after attending the training, teachers increased awareness on the use of alternative disciplines and realised that beating students is not a solution, but they can advise students and understand subjects well (Coordinator, endline)

#### Advising and counselling students

Counselling or providing advice to misbehaving students was reported to be an integral part of changing teaching practice; this strategy was used primarily with students who missed class or refused to follow teachers' instructions. Teachers reported that they used these methods instead of whipping or caning students. One of the headteachers confirmed providing advice:

> When a student has refused to follow what I have ordered to do during the lesson, I just sit down with that student and give him/her advice instead of beating, when seems impossible for him/her to change, I report to the school disciplinary committee for more advice. (Headteacher, midline).

A discipline teacher also testified:

> I have to advise students, for example, let us say a student is out of class or out of school with a boy when the subjects are going on I have to give advice on the

effects of staying out with this boy when it is class time (Discipline teacher, midline)

Teachers described EmpaTeach recommended practices and a change in their behaviour around positive interactions with students:

Now, the only discipline that I give them (students) is through counselling and advising them to change their behaviours. For example, when they have dirty clothes, I tell them to go and change or wash them instead of beating them (Teacher, midline)

These positive perceptions of a change in the use of corporal punishment occurred across both Congolese and Burundian schools. Some students also felt that beating was not common, although it was less clear whether this was related to the intervention itself or not:

Students in this school are not beaten. I think it is not good to beat someone, when one can only advise him or her via a word-of-mouth without beating him or her. Beating students destroys our relationship with teachers (Student, endline)

### Praise and rewards

Both school staff and students discussed the technique of using praise and rewards extensively. Schools developed a system of recognising outstanding students, and teachers included praise practices such as singing positive songs, clapping and announcing outstanding and exemplary teachers before others as an expression of goodwill:

There are changes since praising students is not just necessarily giving them a gift but can also be about something saying, 'well done' and others clapping hands are enough and he or she appreciates such recognition. (Headteacher, midline).

Many teachers from Burundian and Congolese schools mentioned clapping to applaud and encourage good-performing students in class as one of the strategies they employed. Singing a good song to students who did well in class in front of their fellow students brought about a feel-good factor, making it an important motivational approach; this was also reported by the EmpaTeach coordinators. The system of rewarding students for their academic performance also featured at endline, as the participants mentioned clapping, congratulating, appreciating and recognising outstanding students in front of their peers, as reflected in the quote.

When the teacher asks questions in class, the one who raises his/her hand and answers the question correctly is praised by his/her fellow students by clapping of hands. (Student, midline).

### Classroom management practices

Some school staff described finding the classroom management aspects of EmpaTeach useful and listed a range of techniques learnt during the intervention that

they subsequently used in their teaching practice. These included participatory ways of developing rules and regulations, which were mentioned by teachers at midline and endline as a positive technique for managing students inside and outside the classroom who complied with what they proposed to be included in the rules and guidelines.

We were taught some techniques that are positive including the use of non-corporal punishment and learnt how to prepare collaborative rules and regulations in the class and know how to control the classroom, so those are the various benefits that we have gained. (Teacher, endline)

Teachers also saw revising different topics as a positive strategy to keep students engaged in class. Teachers perceived this approach to have led to improvements in understanding among students of the lesson content, hence reducing chances of attracting undue punishment:

Yes, there are changes… before we got training, when you teach a lesson, the students were likely to forget since their brains are not mature enough for the whole lesson to stick in. However, this training has made us learn that if a student fails to respond to the question asked in class as a teacher you cannot just beat him or her instead you make a simple revision to remind the lesson just learned. (Coordinator, midline)

However, these practices were not as widespread as others, such as the use of praise, and not all teachers described using them.

## Teachers' skills development: learning self-regulation and well-being techniques to control emotions and manage stress
### Emotional regulation and well-being techniques

Generally, teachers hailed the EmpaTeach programme for imparting them crucial skills and knowledge on methods and mechanisms for improving their mental health by controlling and managing stress and emotions.

During the training sessions, teachers reflected on their own values, experiences with using corporal punishment and their ability to change. Phrases such as 'EmpaTeach intervention *helps to change or correct teacher's bad behaviour', '(It) intends to alter types of punishments given to schoolchildren', and '(It is) a good programme that provides skills to enable teachers to control/manage their classes'* emerged during interviews with EmpaTeach coordinators.

Teachers and EmpaTeach coordinators described having learnt skills and techniques during the intervention, which helped them to control their emotions and manage stress. They reported praying to be one of the useful strategies before acting when students annoyed them.

When I feel stressed out, I try as much as possible to avoid it through praying. (Coordinator, midline)

However, teachers went further in explaining their ways of controlling emotions, including playing with students,

reading books and walking around outside the class. Others ways that were not directly linked with the intervention included singing songs and listening to music.

> Nowadays we have been given techniques and guidelines to follow to control our emotions, for example, when a student annoys you, you can go outside the class and walk around by the time you return to class your mood will have changed and you feel good and your emotions disappeared (Coordinator, midline)

Some teachers also reported finding these strategies helpful even when outside school, such as at home. Despite the learnt techniques and own ways to control emotions and manage stress, some study participants also described that life hardships were such that it was hard to avoid stress.

> Human beings cannot live without money, and we work to earn at least a little money to get rid of life's hardships. Now the stress comes when you do not have money or (subjected to) low salaries and this is a big problem for me…This training has helped us to change because first when I leave home, I move with only one goal to go and teach students. Our training has focused us not on having other things occupying our minds when teaching. (Coordinator, midline)

### Improving empathy and relationships to support behaviour change

After participating in EmpaTeach, some teachers reported reaching an agreement that, in cases of students' misbehaviour, a discussion should be held between teachers, students and, where necessary, with parents as well, who would be invited to the school for discussions. Both at midline and endline, study participants perceived the approach to have reduced fear and brought students closer to their teachers:

> We do sit with the students and ask them about their problems because sometimes they might be having family problems. We do invite respective parents to come to school to discuss the development of the students. I learn this strategy from the trainings (Coordinator, midline)

Students also reported that teachers were closer to them and listened to their concerns in addition to allocating time to discuss different issues pertaining to school performance and their health status:

> Yes, we do discuss academic issues, for example, when a teacher is teaching and in case there is a part I do not understand, I can raise my hand and say I do not understand, and the teacher will repeat and elaborate more so that I can understand well (Student, midline)

Staff reported that such positive communication allowed teachers to support students with challenges they faced in their lives. A few students also acknowledged

benefiting from such changes. As one of them pointed out:

> I have started seeing them (teachers) changing. Previously, they were treating us badly but nowadays they sit with us and talk to us in a good way (Student, midline)

### Peer and social support in the process of behavioural and attitude change

Participants reported that EmpaTeach sessions allowed teachers to reflect on their strengths and ability to change, which facilitated changes in attitudes and encouraged good practices. Teachers confirmed the sessions to have stimulated changes of norms from their usual practice to 'oneness' or 'togetherness' as they started treating themselves equally and increased co-operation in problem-solving which finally enhanced and sustained good relationships among teachers and students. School staff utilised the skills gained from the EmpaTeach programme to orient peers from the intervention who has missed some of the sessions.

> In the past, teachers had no habit of building each other's capacity. After the training programme they started some group discussions to build each other's capacity. For example, the capacity to teach some of the subjects (Discipline teacher, endline)

This support was deemed a positive move in bringing about the desired changes with implications on students' behaviour and performances.

School staff declared to have received support from their leaders through recognition of their outstanding performance and acknowledgment which boosted their working morale during implementation of EmpaTeach activities. Overall, support from leaders contributed to a positive experience with the EmpaTeach intervention and suggested activities were well received:

> When you pass anywhere people may mention your name, for example, those people whom I have taught may say someone taught us good (EmpaTeach) programme and taught us to behave well. And we realised that in daily life these (teachings) are useful, and wherever you pass people say the programme went well (Coordinator, midline)

### Persisting norms around corporal punishment

Despite these positive experiences with the intervention and useful strategies learnt, teachers continued to support the use of corporal punishment. For example, some teachers insisted on using such corporal punishment based on their past experiences:

> If a child is punished without using a stick or beaten, he/she tends not to listen. As I said when I was studying, we were punished with sticks, but I never heard of anyone who died because of being beaten using sticks. I am convinced to continue the behaviour

because children of nowadays do not listen to advice…. (Discipline teacher, midline).

Teachers also feared that some students had lost respect for their fellows and claimed their behaviour had become less manageable without corporal punishment. Some students were reported to be lazy or aggressive and rude to the teachers because they knew that even if they misbehaved, they would not be whipped or caned:

School children now do not fear anymore after learning that it is their right to stay free from whips. Some even say If you beat me, I will report you to the law enforcing authorities. (Discipline teacher, midline)

Teachers felt that students would refuse to follow their instructions which was one of the reasons behind the desire to continue using corporal punishment. This perception generally emerged more often in Congolese than in Burundian schools. Students' misbehaviour and attitudes made it difficult for some teachers to enter the class without a cane as testified below.

For example, there are three students making noises while teaching. You order one student to stand up and shift to another chair, but he refuses to shift and tells you that he will not shift. Therefore, I cannot advise a student like that; instead, I cane him even if the programme does not allow us to use corporal punishment. There are types of misbehaviours which when they exceed the (tolerance) limit, I cannot tolerate [them. Any student who goes beyond the teachers' instructions will get strokes of the cane. That is why I am saying that I do not accept other disciplinary methods (Discipline teacher, endline)

Some EmpaTeach coordinators reported that parents also supported corporal punishment as they believed it to be a good way of disciplining students.

It is from parents who said that if you don't beat your child, you discipline him/her in a bad way and will not understand what you are telling him. (Coordinator, midline).

Teachers reported that using non-corporal punishment was perceived by parents as an insufficient way of disciplining students who would not understand what was being taught to them without use of violence.

## DISCUSSION

The study findings revealed how behavioural change interventions can facilitate change among schoolteachers in disciplining students. Teachers declared to have attended several EmpaTeach sessions facilitated by group coordinators and acknowledged the positive role that the intervention played in influencing the way they disciplined students and behaved in class. Teachers reported using a variety of alternative discipline strategies learnt during the sessions including teaching practices to keep students focused, reward and praise methods to reinforce positive student behaviours. They also perceived the techniques for managing stress and controlling their own emotions as beneficial in facilitating avoidance of corporal punishment. Although we know from our trial analyses[15] that the intervention overall did not lead to a reduction in students' experiences of corporal punishment, this process evaluation shows that, school staff felt that there had been positive changes in their disciplinary approaches, including their relationships and communication with students.

Students' responses also depicted a generally positive picture, although their experiences were less consistent and less clearly linked to the intervention itself. These findings are generally consistent with our previous study results;[15] specific techniques were not consistently mentioned and teachers' accounts of new strategies were sometimes vague and not always linked to EmpaTeach content. This suggests that, while the intervention may have fostered improvements in the school environment among school staff, it was not successful at consistently developing teachers' skills along all the intervention components.

The study offers important insights into student and teacher experiences of the intervention and represents a first attempt to reflect on the implementation and pathways of a school-based violence prevention intervention in a fragile setting. Data collection was delayed by COVID-19 restrictions in the camp which may have affected participants' ability to respond adequately to all sections of the interview due to a lack of continuity in practising what they learnt in the intervention sessions. It is important to note that while the EmpaTeach intervention helped to change some teachers' behaviour and attitudes towards the use of corporal punishment, during the implementation of the intervention there were challenges. Difficulties in balancing coordination responsibilities alongside teaching and the rainy season, which interfered with timely and effective delivery of intervention activities, may have prevented teachers from fully engaging with the intervention content.

Lack of external continued support and supervision may have contributed to lack of clarity of certain strategies and compromised the internalisation of specific techniques. This study plays a crucial role in intervention evaluations by providing detailed information on how the intervention was perceived and experienced. Through in-depth exploration, contextual understanding and ability to uncover underlying mechanisms, qualitative research offers valuable insights that quantitative methods alone cannot produce. Additionally, conducting research with the support of research assistants who shared the cultural and linguistic background of the participants greatly enhanced the quality of the data collected during the evaluation, ensuring smooth communication and facilitating the acquisition of rich data. In this process evaluation of the EmpaTeach intervention, valuable insights were obtained by directly engaging with stakeholders,

including teachers and students who were directly involved in the environments where violence may occur.

However, the study counts some limitations as it recruited a small sample size that prevents the generalisation of study findings. Finally, recruitment or replacement of new study participants at midline and endline made it difficult to capture continuity in the stories as related to the EmpaTeach intervention. The study faced additional limitations, such as an incomplete exploration of the intervention theory and contextual factors in interviews with students and teachers, which hindered a comprehensive understanding. Additionally, reliance on participants' subjective accounts restricted our ability to assess all potential mechanisms of action within the interventions.

Despite the limitations faced during the intervention implementation, our study found that support and facilitation by peers, and the self-guided nature of the intervention package may have contributed to the positive responses of school staff. Whereas the quantitative findings stemming from the trial showed that there was no evidence that students' reports of physical violence from teachers decrease on average, this qualitative study showed that overall teachers perceived the intervention as useful to improve their teaching abilities and reduce their use of corporal punishment. This contrast in findings might be attributable to several possibilities. To begin with, our trial outcome was the calculated as the proportion of students reporting any experience of corporal punishment in the past week, not accounting for frequency or severity. Second, there is some suggestion in our qualitative data of improved relationships between teachers and students and a more positive and friendly classroom environment; it is possible that some teachers did use less violence, or less severe violence, without stopping. Other studies have shown that participants in Uganda, for example, understand 'corporal punishment' to refer to only excessive beating.[10]

In reflection, EmpaTeach is different from other successful violence prevention interventions in its focus on stress reduction and prevention of impulsive violence, and because it does not explicitly seek to confront teachers on the use of corporal punishment. Instead, the intervention seeks to enhance teachers' repertoire of techniques for classroom management and intended for these to replace the use of corporal punishment. Other interventions which have some evidence of reductions in corporal punishment are thought to have different mechanisms of effect. The ICC-T intervention, implemented in Tanzania, provided training on teacher–student interaction and guided teachers to reflect on child maltreatment. The intervention was found to have resulted[11] in a significant change in the use of physical and emotional violence among teachers to students in intervention schools, as reported by both teachers and students. The Irie Classroom Toolbox intervention was implemented in Jamaican preschools through the provision of training to teachers in classroom behaviour management and

focused on creating an emotionally supportive classroom environment; preventing and managing child misbehaviour; strengthening teacher social and emotional skills; promoting individual and class-wide behaviour planning.[7] In Uganda, the GST intervention aimed to change the schools' operational cultures through the involvement of the school administration, teachers and students and surrounding communities. The GST's main mechanisms of action appeared to relate to improved teacher–student relationships, which then improved students' voices, reduced fear through rewards and praise, and encouraged good behaviour.[10] Similar changes have been noted in the EmpaTeach intervention whereby the reward and praise were perceived by school staff as a positive mechanism for fostering behavioural change among students. However, teachers perceived parents to be supportive of the use of corporal punishment, as reported in other settings.[4 19] Attitudes supportive of corporal punishment and old practices continued to coexist alongside new intervention strategies that were implemented.

Our data also highlight the variation in use of corporal punishment across schools, and in what teachers perceived as factors responsible for such practice. In some schools, teachers saw students' reluctance to change or obey to the instructions given, and parents' belief that beating is the best way to discipline students, as the main reasons why corporal punishment continued. Teachers also perceived that the use of alternative discipline would only be effective for certain students, and 'lazy' and 'misbehaved' students were thought to only respond to corporal punishment. Similar perceptions were reported in other contexts.[20 21] As such, there is a need for further contextual analysis of what is perceived as violence, whose findings will inform specific strategies for intervention development. For future testing of the intervention, there is a need to critically review the components of EmpaTeach including the sessions content, delivery modality, contextual characteristics and duration.

## CONCLUSION

The EmpaTeach intervention to reduce physical violence from teachers to students was widely accepted by teachers in Nyarugusu refugee camp. Some intervention techniques, especially alternative discipline practices and classroom management strategies, were widely implemented, and teachers were familiar with stress reduction and emotional regulation methods. However, it appears that the norms sustaining the use of corporal punishment persisted among teachers as they kept referring to their past personal experiences of school violence from when they were studying and to the belief that physical violence is the best way to discipline students. These qualitative findings revealed high acceptability among teachers and coordinators of the alternative disciplinary and classroom management methods included in EmpaTeach, thus we recommend for the intervention to be further adapted and tested, potentially also in non-humanitarian settings.

**Acknowledgements** We are grateful to the UK Medical Research Council for financial support. Thanks are extended to Mary Qiu, Martin Zuakulu and Dennis Nombo for supervising data collection; Sia Joseph, Fortunata Kirita, Gloria Angolo and research assistants recruited in the Camp for their assistance at different stages of the data collection, translation and coding process and IRC for logistical support during the data collection process. Study participants including the EmpaTeach Coordinators, teachers and students are acknowledged for consenting to participate in the study.

**Contributors** MB: collected and analysed the data, drafted the manuscript, revised and finalised for submission. EHS and VB: designed the study, supervised data collection, analysed the data, supported MB to draft the manuscript and in several rounds of revisions. ZK: collected the data, coding and reviewed the manuscript. ET, EE, CF and KR: contributed to writing and revised several versions of the manuscript. GMM: revised the manuscript. KD: obtained funding, designed and supervised all aspects of the study, reviewed manuscript and was the guarantor of the study. All authors have read and approved the manuscript.

**Funding** The research was supported by funding from an anonymous donor to the International Rescue Committee and by the Medical Research Council of UK with reference number MRC/DfID/NIHR MR/S023860/1 to Karen Devries at the London School of Hygiene and Tropical Medicine.

**Competing interests** Katherine Rodrigues was involved in the development of th EmpaTeach intervention at the International Rescue Committee

**Patient and public involvement** Patients and/or the public were not involved in the design, or conduct, or reporting or dissemination plans of this research.

**Patient consent for publication** Not required.

**Ethics approval** This study involves human participants and was approved by Medical Research Coordinating Committee of the National Institute for Medical Research (NIMR) in Tanzania (with reference number NIMR/HQ/R.8a/Vol. IX/2920) and the London School of Hygiene Ethics Committee in the UK (16000-1). Participants gave informed consent to participate in the study before taking part.

**Provenance and peer review** Not commissioned; externally peer reviewed.

**Data availability statement** Data are available upon reasonable request. Fully anonymised quantitative data are available on request from the LSHTM Data Repository (https://doi.org/10.17037/DATA.0002474) for researchers who meet the criteria for data access and whose intended analyses fall under the scope of the PVAC study. The LSHTM Research Data Manager, based in the Library and Archives Service, is responsible for managing data in the repository and can be contacted at researchdatamanagement@lshtm.ac.uk.

**ORCID iDs**
Elizabeth H Shayo http://orcid.org/0000-0001-9131-1784
Karen Devries http://orcid.org/0000-0001-8935-2181

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
