## [Reviewer comments · BMJ Open]

ARTICLE DETAILS

TITLE (PROVISIONAL)	A qualitative process evaluation of the EmpaTeach intervention to reduce teacher violence in schools in Nyarugusu Refugee Camp, Tanzania
AUTHORS	Bakari, Mtumwa; Shayo, Elizabeth; Barongo, Vivien; Kiwale, Zenais; Fabbri, Camilla; Turner, Ellen; Eldred, Emily; Mubyazi, Godfrey M.; Rodrigues, Katherine; Devries, Karen

VERSION 1 – REVIEW

REVIEWER	Njelesani, Janet New York University, Occupational Therapy
REVIEW RETURNED	06-Feb-2023

GENERAL COMMENTS	Thank you for your work on a very important topic that to date has not received significant attention in the literature. The included quotes illuminate the responses of both teachers and students, which is an important contribution of the work as students' experiences are often not considered nor collected. To center these perspectives, I suggest embedding all of the quotes within the findings section instead of having a separate table. The majority of issues noted in the paper noted stem from a lack of congruency between the chosen methodology (i.e., qualitative) and the study sections. Including a section about the theoretical framework that guided the study would strengthen the reader's ability to appraise the methodological congruence across study sections and better understand how methods were developed and analysis was conducted. I appreciate that components of reflexivity have been included for the researchers conducting data collection but reflexivity is not indicated for the rest of the research team, including importantly for those conducting analysis. The analysis section is not clear in the abstract or the main body of the text. It is written that thematic analysis was conducted but no reference was included and the written description is closer to content analysis. It is stated that data saturation was achieved; however, more details are needed on how saturation was determined given the small sample of participants in each participant group. The discussion section would benefit from more details. First, how do these qualitative findings relate to the wider literature on teacher violence against students? Next, much of the discussion is supported by assumptions made by the authors but is not theoretically supported with accompanying references, in particular the section on why teachers felt a positive change had occurred. Furthermore, an unpacking of how the findings differed across participants from the two countries (e.g., the topic of corporal punishment emerged more for Congolese teachers) would also strengthen the discussion section. More details are also needed on how would the intervention be modified given the
--

	lack of effectiveness before being scaled up. The limitations noted are written from a positivist positionality. Limitations for this study could instead include examples such as child-friendly methods were not used to generate data with child participants therefore the depth of data collected may be limited. Another limitation could be the lack of attention to the intersectionality of identities as the study did not appear to purposively consider findings from a gender perspective or include children with disabilities, etc. A moderate amount of grammatical errors remain throughout the paper (e.g., capitalized words in the middle of sentences, no period at end of sentences, one-sentence paragraphs, etc.).
--	---

VERSION 1 – AUTHOR RESPONSE

1	The included quotes illuminate the responses of both teachers and students, which is an important contribution of the work as students' experiences are often not considered nor collected. To centre these perspectives, I suggest embedding all of the quotes within the findings section instead of having a separate table.	All quotes have been inserted within the respective finding's sections. Please see pg 12 to 21
2	The majority of issues noted in the paper noted stem from a lack of congruency between the chosen methodology (i.e., qualitative) and the study sections. Including a section about the theoretical framework that guided the study would strengthen the reader's ability to appraise the methodological congruence across study sections and better understand how methods were developed and analysis was conducted.	The study has relied on the theory of change connected to Empa Teach intervention. Some more information is included (pg 6 ad 7) Also, we used thematic analysis
3	It is written that thematic analysis was conducted but no reference was included and the written description is closer to content analysis.	Thematic analysis was used during data analysis by identifying themes and patterns and (Brown T. 2006) used as a reference for confirmatory factor analysis. Pg 10
4	It is stated that data saturation was achieved; however, more details are needed on how saturation was determined given the small sample of participants in each participant group.	Saturation level achieved when participants asked questions and there were no more new information coming out during the interview. See pg 10
5	how do these qualitative findings relate to the wider literature on teacher violence against students? Next, much of the discussion is supported by assumptions made by the authors but is not theoretically supported	The discussion section revised to compare with other intervention on behavioural change implemented other countries. Pg 27-28

	with accompanying references, in particular the section on why teachers felt a positive change had occurred.	
6	An unpacking of how the findings differed across participants from the two countries (e.g., the topic of corporal punishment emerged more for Congolese teachers) would also strengthen the discussion section. More details are also needed on how the intervention would be modified given the lack of effectiveness before being scaled up.	The reason for persistence of corporal punishment shown in the discussion section. Pg 26 More explanations for things to consider before scaling up the intervention are included in pg 29.
7	A moderate amount of grammatical errors remain throughout the paper (e.g., capitalized words in the middle of sentences, no period at end of sentences, one-sentence paragraphs, etc.).	Thank you. Changes have been effected through out the manuscript. Some capitals are in special words such as Empa Teach etc

VERSION 2 – REVIEW

REVIEWER	Njelesani, Janet New York University, Occupational Therapy
REVIEW RETURNED	12-Jun-2023
GENERAL COMMENTS	The revisions have strengthened the paper, in particular the discussion section. Before publication, another proofread of the text is required as significant grammatical errors remain.